# Significant Enhancement of 5-Hydroxymethylfural Productivity from *D*-Fructose with SG(SiO_2_) in Betaine:Glycerol–Water for Efficient Synthesis of Biobased 5-(Hydroxymethyl)furfurylamine

**DOI:** 10.3390/molecules27185748

**Published:** 2022-09-06

**Authors:** Daozhu Xu, Qi Li, Jiacheng Ni, Yucai He, Cuiluan Ma

**Affiliations:** 1School of Pharmacy, Changzhou University, Changzhou 213164, China; 2State Key Laboratory of Biocatalysis and Enzyme Engineering, Hubei University, Wuhan 430062, China

**Keywords:** *D*-fructose, 5-Hydroxymethylfural, 5-(Hydroxymethyl)furfurylamine

## Abstract

5-Hydroxymethyl-2-furfurylamine (5-HMFA) as an important 5-HMF derivative has been widely utilized in the manufacture of diuretics, antihypertensive drugs, preservatives and curing agents. In this work, an efficient chemoenzymatic route was constructed for producing 5-(hydroxymethyl)furfurylamine (5-HMFA) from biobased *D*-fructose in deep eutectic solvent Betaine:Glycerol–water. The introduction of Betaine:Glycerol could greatly promote the dehydration of *D*-fructose to 5-HMF and inhibit the secondary decomposition reactions of 5-HMF, compared with a single aqueous phase. *D*-Fructose (200 mM) could be catalyzed to 5-HMF (183.4 mM) at 91.7% yield by SG(SiO_2_) (3 wt%) after 90 min in Betaine:Glycerol (20 wt%), and at 150 °C. *E. coli* AT exhibited excellent bio-transamination activity to aminate 5-HMF into 5-HMFA at 35 °C and pH 7.5. After 24 h, *D*-fructose-derived 5-HMF (165.4 mM) was converted to 5-HMFA (155.7 mM) in 94.1% yield with *D*-Ala (*D*-Ala-to-5-HMF molar ratio 15:1) in Betaine:Glycerol (20 wt%) without removal of SG(SiO_2_), achieving a productivity of 0.61 g 5-HMFA/(g substrate *D*-fructose). Chemoenzymatic valorization of *D*-fructose with SG(SiO_2_) and *E. coli* AT was established for sustainable production of 5-HMFA, which has potential application.

## 1. Introduction

With the continuous growth in population and the high dependence on non-renewable resources, it is a vital step to switch to alternative clean and renewable energies [1]. Among varied renewable resources, lignocellulosic biomass (LCB) continues to attract much attention as an inexpensive and renewable alternative to fossil fuel to manufacture biofuel molecules and biobased compounds [2]. Biomass has been widely used as feedstock to produce a series of valuable compounds, such as biofuels, biopolymers, energy-rich chemicals, and bioactive molecules [3]. Consequently, there has been a growing interest in the cost-effective valorization of mono- and poly-saccharides into value-added platform compounds in biorefinery processes [4].

5-Hydroxymethylfural (5-HMF), which can be prepared via the dehydration of biomass-derived monosaccharide (e.g., *D*-fructose), is one of the foremost promising platform chemicals [5]. Generally, a number of homogeneous and heterogeneous acid catalysts widely catalyze dehydration of carbohydrates to generate 5-HMF [6]. Recently, the preparation of 5-HMF from carbohydrates has mostly employed corrosive inorganic acids (e.g., HCl, H_2_SO_4_, etc.), preparation of complicated and expensive solid acids, and non-environment friendly metal catalysts [7]. Hence, it is urgent to explore new sustainable catalysts that are in line with the concept of green chemistry to manufacture 5-HMF from carbohydrates [8].

5-HMF has strong reactivity, due to the presence of three functional groups (a furan ring, hydroxy group and an aldehyde group), which can be further valorized into a series of value-added furans [9,10]. 5-(Hydroxymethyl)furfurylamine (5-HMFA), as an important 5-HMF derivative, is utilized for manufacturing drugs, but can also be used as a curing agent for epoxy resins [11]. Typically, 5-HMFA has been mainly synthesized by means of a chemo-catalysis route [5], which usually employs inexpensive, or unfriendly, metal catalysts. In addition, chemo-catalysis can suffer from energetic consumption issues under the harsh performance conditions (e.g., high pressure, high temperature, etc.) [12]. Very recently, bio-catalysis has gained great interest to prepare 5-HMFA, due to the mild reaction conditions, good catalytic activity, excellent selectivity, and eco-friendliness [13].

The strategy for bridging nonenzymatic and enzymatic catalysis has been well established for the production of value-added building-blocks [14,15]. Chemoenzymatic cascade catalysis enables two or more steps in a one-pot manner, which can ignore the separation of intermediates and, thus, reduce performance time and performance cost [16]. In chemoenzymatic conversion, the reaction medium has a vital role in influencing catalytic efficiency. Water is known to be a typical green solvent for preparing 5-HMF and its derivatives [17]. As a polar protic solvent, water can easily cause the occurrence of side-reactions and low productivity of 5-HMF [18]. The past decade has witnessed the expeditious development of deep eutectic solvents (DESs) and their wide applications in chemo-catalysis and bio-catalysis reactions [19,20]. DESs have many advantages, such as extremely low toxicity, good biodegradability, and biocompatibility with enzymes [21,22]. Combining biomass pretreatment and whole cell catalysis for preparation of 5-HMFA in a DES–water medium is a promising strategy for sustainable production of biobased compounds from biomass.

Herein, a hybrid synthetic route for producing 5-HMFA from biobased *D*-fructose was constructed in a tandem reaction with SG(SiO_2_) chemo-catalyst and ω-transaminase biocatalyst in a betaine-based DES–water medium (Figure 1). The effect of medium composition, dehydration temperature and duration, catalyst SG(SiO_2_) dose, and *D*-fructose dosage were examined in terms of the dehydration of *D*-fructose into 5-HMF with SG(SiO_2_) chemo-catalyst. Subsequently, recombinant *E. coli* AT containing ω-transaminase [23] was employed to aminate *D*-fructose-derived 5-HMF into HMFA in a DES–water system. Finally, a hybrid reaction with SG(SiO_2_) chemo-catalyst and *E. coli* AT biocatalyst was conducted for sustainable preparation of 5-HMFA from biobased *D*-fructose in eco-friendly DES–water.

## 2. Results and Discussion

### 2.1. Investigation of B:Gly–Water Composition on the 5-HMF Formation

The yield of chemical and biological reactions is largely dependent on the reaction medium [24]. Figure 2a showed the results of SG(SiO_2_) solid acid catalyzing *D*-fructose dehydration to form 5-HMF at 150 °C in various B:Gly–water media. In the aqueous system without DES, the high yield of 5-HMF was only 1.7%. In a certain range of B:Gly–water ratio, the 5-HMF yield was augmented when the B:Gly-to–water mass ratio became higher. After dehydration for 1 h, the 5-HMF yield reached 69.7% [in B:Gly (5 wt%)], 74.0% [in B:Gly (10 wt%)], 76.4% [in B:Gly (15 wt%)], 76.8% [in B:Gly (20 wt%)], and 82.6% [in B:Gly (25 wt%)], implying that the addition of an appropriate amount of B:Gly was favorable for 5-HMF formation. The reason might be that B:Gly might confine the degradation of 5-HMF. During the *D*-fructose dehydration, the yield of 5-HMF showed different trends when the B:Gly–water ratios were different. In the presence of 25 wt% B:Gly, the 5-HMF yield continued to decrease with the increase of dehydration time. At other ratios of B:Gly to water (5:95, 10:90, 15:85, and 20:80; wt:wt), the 5-HMF yield increased first but, then, gradually decreased, reaching the maximum yield of 78.3%, 81.5%, 85.3%, and 91.7%, respectively. *D*-Fructose was dehydrated into 5-HMF in a yield of 78% by Amberlyst 15 chemo-catalyst in ChCl:GA at 60 °C for 4 h [25], and other catalysts and reaction solvents were used to synthesize 5-HMF. WCl6 catalyst converted *D*-fructose into 5-HMF in a 72% yield in THF–[Bmim]Cl medium at 50 °C for 4 h [6]. [C_3_SO_3_Hmim][HSO_4_] could dehydrate *D*-fructose to 5-HMF with a yield of 73% in MIBK–H_2_O, respectively [4]. 5-HMF yield reached 35% by co-catalysis with malic acid and Betaine/HCl at 140 °C for 11 min in H_2_O–ethyl acetate [7]. NbPO dehydrated *D*-fructose to 5-HMF with a yield of 70% [8]. In this work, the maximum yield of 5-HMF was only 1.7% in the aqueous system containing SG(SiO_2_) (Appendix A). In the B:Gly–water system without SG(SiO_2_), the highest yield of 5-HMF was 48.7%. In B:Gly–water (B:Gly, 20 wt%) containing SG(SiO_2_) solid acid catalyst, the 5-HMF yield reached a maximum of 91.7% at 150 °C for 1.5 h. The main reason was that the medium containing B:Gly affected the hydrogen-bond network, and co-catalysis with B:Gly and SG(SiO_2_) could efficiently catalyze the 5-HMF formation from *D*-fructose. Another reason was that many side-reactions would happen in single water system, while 5-HMF was more stable in B:Gly–water.

### 2.2. Effects of Dehydration Temperature and Duration on the 5-HMF Formation

The effect of reaction temperature and duration on the generation of 5-HMF is presented in Figure 2b. After dehydration for 1–2.5 h, the conversions were 100%. The yield of 5-HMF was only 44.5% at 140 °C after this dehydration reaction proceeded for 1 h, whereas the 5-HMF yield jumped up to 77.1% after catalyzing *D*-fructose reacted for 2.5 h. When the dehydration temperature was increased further, the yield of 5-HMF clearly rose to over 80% in a short period of dehydration time. The highest 5-HMF yield at increasing temperatures could reach 81.4% (160 °C, 0.5 h) and 91.7% (150 °C, 1.5 h). These results indicated that the reaction temperature had considerable effects on the generation of 5-HMF and that high temperatures could shorten the time required for the 5-HMF yield to reach its maximum [1]. Although the higher dehydration temperature shortened the duration for the *D*-fructose dehydration to 5-HMF, the secondary reaction against 5-HMF also caused a decrease in 5-HMF yield [26]. Consequently, the optimal dehydration condition for *D*-fructose dehydration to 5-HMF with SG(SiO_2_) catalyst was carried out in B:Gly–water after 1.5 h at 150 °C.

### 2.3. Investigation of SG(SiO_2_) Dose on the 5-HMF Formation

Figure 3 depicts the effect of catalyst dosage on the 5-HMF yields after the *D*-fructose dehydration reacted at 150 °C for 90 min in B:Gly–water (B:Gly, 20 wt%) system. It had been reported that DESs could convert *D*-fructose to produce 5-HMF as chemo-catalysts and reaction solvents [27]. Without the addition of SG(SiO_2_) to B:Gly–water (B:Gly, 20 wt%) for 90 min at 150 °C, the formed HMF reached a yield of 48.7%. In the presence of a low dose of SG(SiO_2_) catalyst (2 wt%), the productivity of 5-HMF was obviously enhanced, and the yield of 5-HMF reached 71.2%. This might be the role of SG(SiO_2_) solid acid and B:Gly together in catalyzing the generation of 5-HMF from *D*-fructose. As the catalyst SG(SiO_2_) dosage rose from 0 wt% to 3 wt%, the 5-HMF yield obviously increased. Upon raising the dosage of SG(SiO_2_) from 3 wt% to 5 wt%, the 5-HMF yield dropped. The highest 5-HMF yield of 91.7% was achieved using 3 wt% of SG(SiO_2_) as catalyst. *D*-Fructose was transformed to 5-HMF in a dehydration reaction system of DESs (ChCl-phenol)–water with SACS catalyst at 110 °C, achieving a 67% yield of 5-HMF in 4 h [28]. The combination of ChCl:GA (DES) and Amberlyst 15 (catalyst) could dehydrate *D*-fructose to 5-HMF (78% yield) in an aqueous solvent [25]. The above results illustrated that the combination of SG(SiO_2_) solid acid catalyst and B:Gly–water solvent system offered a broad perspective for the economical large-scale production of 5-HMF from *D*-fructose via a dehydration reaction.

### 2.4. Investigation of D-Fructose Dosage on the 5-HMF Formation

From a practical application point of view, the initial concentration of *D*-fructose is an important aspect to the industrial production of, and economic benefits of, 5-HMF [29]. Hence, different *D*-fructose dosages were applied to the synthesis of 5-HMF (Figure 4). It was clearly ascertained that different initial doses of *D*-fructose had perceptible effects on the 5-HMF yield. When the amount of *D*-fructose increased from 0.2 M to 2 M, the corresponding 5-HMF yield decreased from 91.7% to 45.9%, indicating that *D*-fructose dosage could affect the 5-HMF formation. The reason for this phenomenon might be that the degradation products of *D*-fructose further contributed to the degradation of 5-HMF during the whole reaction [29].

### 2.5. The Recyclability of the Catalytic System

The regeneration and reuse of the catalytic system can decrease economic burden and reduce environmental pollution, which is the key to the industrial production of 5-HMF, and has high application potential [30,31]. As presented in Figure 5, the catalytic system was recycled six times in order to test the recyclability of the catalytic system to catalyze the synthesis of 5-HMF from *D*-fructose. 5-HMF was extracted from 5-HMF liquor four times using ethyl acetate by mixing in equal volume. The B:Gly–water medium containing solid acid SG(SiO_2_) was reused for further production of 5-HMF. After the first round, the 5-HMF yield in the catalytic system was 91.7%, and once the three rounds of reaction were over, the 5-HMF yield in the catalytic system decreased slightly to 83.1%. When *D*-fructose was catalyzed from 3rd to 6th batch, 5-HMF yields decreased from 83.1% to 72.4%. The inevitable accumulation of humic substances, could be why the yield of 5-HMF began to decline. In the 6th batch, the catalytic system also efficiently catalyzed *D*-fructose into 5-HMF. When the CrCl_3_-[N_2222_]Cl/EG catalytic system was reused from 1st to 4th batch, the 5-HMF yield dropped from 42% to 38% [32]. These results indicated that the SG(SiO_2_) catalyst with B:Gly–water could be efficiently recycled for *D*-fructose dehydration to produce 5-HMF.

### 2.6. Proposed Catalytic Mechanism for Dehydration of D-Fructose

A proposed mechanism for SG(SiO_2_)-catalyzed dehydration of *D*-fructose to 5-HMF in a B:Gly–water medium is presented in Figure 6. When fructofuranoses were formed in the reaction medium, the hydroxyl groups on the saturated carbon were rapidly protonated [33]. Due to the presence of H^+^, Cl^−^ and silanol groups, the reaction for removal of the first water molecule would be more effective, which allowed good formation of the enediol intermediates. Upon the action of hydrogen-bonding on C-H and C-O bonds and/or electrostatic forces, the intermediates could facilitate the removal of two water molecules to generate 5-HMF. Electrostatic effects and hydrogen bonding on the polar silanol-rich surface of silica [34], would weaken the stability of C-H bonds and hydroxyl groups in *D*-fructose [35]. This effect might promote the *D*-fructose dehydration by using B:Gly with strong electron-withdrawing ability as the reaction medium. The presence of B:Gly might be beneficial for accepting and donating electrons to improve dehydration of *D*-fructose. Significantly, B:Gly had a great promoting effect on the *D*-fructose dehydration catalyzed with SG(SiO_2_). This effect of hydrogen bonding within the DESs system ultimately improved the *D*-fructose dehydration to 5-HMF.

### 2.7. Biotransformation of 5-HMF into 5-HMFA

Bio-catalysis is used to produce high-value chemicals, due to its gentle reaction conditions, few side-reactions and high efficiency of production [36]. Alanine has been considered the most widely used amine donor. In this study, *D*-Ala was used as amine donors during the bio-amination of 5-HMF into 5-HMFA with *E. coli* AT whole cells. Increase of the amine donor dose in the reaction process could be used to alter the reaction balance to generate the product of interest [37]. Figure 7a shows the effect of *D*-Ala dosages on bio-transamination. By increasing the *D*-Ala to 5-HMF molar ratio from 2:1 to 15:1 at 35 °C and pH 7.5 in B:Gly–water (B:Gly, 20 wt%), bio-transamination activity and 5-HMFA selectivity were increased. *D*-Ala and 5-HMF molar ratio of 15:1 was found to provide the optimal reaction conditions. Further increasing the dosages of *D*-Ala in the reaction led to a slight decrease in the bio-catalytic activity and 5-HMFA selectivity, indicating that potential inhibition occurred when the molar ratio of *D*-Ala-to-5-HMF exceeded 15:1 in the bio-transamination reaction.

The tolerance of enzymes to high substrate concentrations is a major process parameter in synthetic biology for the development of efficient biocatalysts [38]. Hence, the substrate tolerance of recombinant *E. coli* AT was investigated under seven different dose of 5-HMF in B:Gly–water (Figure 7b). Interestingly, the selectivity of the product 5-HMFA was low when the substrate 5-HMF dose was low. With the continuous increase of the 5-HMF dose, the selectivity of 5-HMFA also increased. *E. coli* AT could still catalyze the generation of 5-HMFA with high efficiency as the 5-HMF dosage was further increased to 200 mM. By raising the concentration of 5-HMF from 50 mM to 200 mM, the yield of 5-HMFA increased gradually. When the concentration of 5-HMF increased from 200 mM to 500 mM, the yield of 5-HMFA dropped from 98.3% to 52.0%, while the selectivity of bio-amination remained at a high level. Raney Ni was used to chemically catalyze 5-HMF into 5-HMFA (80.7%, yield) under harsh reaction conditions (0.35 MPa NH_3_, 120 °C, and 1 MPa H_2_) [5]. To sum up, *E. coli* AT was able to aminate 5-HMF (200 mM) with *D*-Ala (*D*-Ala-to-5-HMF molar ratio of 15:1) into 5-HMFA with a yield of 98.3% in B:Gly–water (B:Gly 20 wt%, 35 °C, and pH 7.5). Distinct from chemo-catalysis, bio-catalytic synthesis of 5-HMFA from 5-HMF had milder performance conditions, higher yield and easier operation, which was a green and sustainable synthetic route.

### 2.8. Chemoenzymatic Catalysis of D-Fructose to 5-HMFA in B:Gly–Water

Chemoenzymatic cascade catalysis, which combines the unique advantages of both non-enzymatic and bio-catalytic reactions, namely the reactivity of chemical catalysts and the high selectivity of enzymes, has become an emerging strategy of interest [38,39]. As illustrated in Figure 8, a combination of chemo-catalysis, using SG(SiO_2_) solid acid, and bio-catalysis, using *E. coli* AT, was used to tandemly convert *D*-fructose into 5-HMFA. An amount of 50 mL B:Gly–water (B:Gly, 20 wt%) containing SG(SiO_2_) (3 wt%) catalyzed 200 mM *D*-fructose at 150 °C to give 183.4 mM 5-HMF after 90 min. The formed 5-HMF liquor was adjusted to pH 7.5 and, then, *E. coli* AT (0.050 g/mL) and *D*-Ala (*D*-Ala-to-5-HMF molar ratio 15:1) were supplemented for the amination of 5-HMF to generate 5-HMFA. Bio-transamination at 35 °C for 24 h, resulted in 165.4 mM of 5-HMF being transformed into 5-HMFA (155.7 mM) in a yield of 94.1%. 5-HMFA ^1^H NMR (CD_3_OD, 400 MHz): δ 1.89 (s, 2H, NH_2_), 3.94 (s, 2H, CH_2_NH_2_), 4.50 (s, 2H, CH_2_OH), 6.27–6.28 (d, *J* = 4.0 Hz, 1H, furan H), 6.32–6.33 (d, *J* = 4.0 Hz, 1H, furan H). Overall, 200 mM *D*-fructose could be converted to 155.7 mM 5-HMFA via a chemoenzymatic approach.

A mass balance was calculated from *D*-fructose to 5-HMFA in a B:Gly–water system (Figure 9). *D*-fructose (1.8 kg) was dehydrated with SG(SiO_2_) (1.5 kg) and DESs B:Gly (10 kg) under a temperature of 150 °C for 90 min. The resulting 5-HMF liquid containing 1.16 kg of 5-HMF was adjusted to pH 7.5 and then added to *E. coli* AT (2.77 kg) and *D*-Ala (12.25 kg) for biological amination at 35 °C, achieving the productivity of 0.61 kg 5-HMFA/kg substrate *D*-fructose at pH 7.5 in 1 d. Coupling of a biocompatible non-enzymatic catalyst and a highly selective enzymatic catalyst to carry out the dehydration of *D*-fructose into 5-HMFA in B:Gly–water was feasible, which would diminish equipment input, and avoid the separation of intermediates.

Chemo-catalysis and bio-catalysis approaches have been used in the synthesis of 5-HMFA [5,40]. 5-HMFA formed in a 4.3% yield from furfurylamine with HCl (6 M) in a formaldehyde solution (35 wt%) at 30 °C after 15 min [40]. Furfurylamine was aminated to 5-HMFA (15% yield) by Amberlyst-15 and formalin at 40 °C after 1 h [41]. Ru/C, Pd/C and Pt/C could catalyze 5-HMF into 5-HMFA with yields of 40.5%, 57.5% and 81.2% under high pressure, respectively [5]. The production of 5-HMFA by the chemo-catalysis technique suffered from several drawbacks, such as unfriendliness, low yield, poor selectivity and harsh performance conditions. In contrast, bio-catalysis has many advantages, such as mild reaction conditions, eco-friendliness and excellent selectivity. The valorization of monosaccharides into 5-HMFA, using chemoenzymatic cascade catalysis, bridges chemical and biological catalysis. In this work, an efficient conversion of biobased *D*-fructose to produce 5-HMF was carried out using SG(SiO_2_) solid acid catalyst in B:Gly–water. During the process of preparing 5-HMF, the reaction medium B:Gly–water played a pivotal role. Relative to the aqueous phase, the addition of DESs could greatly promote the 5-HMF yield. Sequentially, *E. coli* AT cell harboring ω-transaminase was able to aminate *D*-fructose-derived 5-HMF to produce 5-HMFA by using *D*-Ala as amine donor under ambient conditions, accompanied with the formation of pyruvate (Figure 10). To synthesize 5-HMFA, one-pot catalysis of *D*-fructose was carried out by tandem conversion with SG(SiO_2_) and *E. coli* AT cell in B:Gly–water, achieving a remarkable 5-HMFA yield of 86.3% (based on *D*-fructose). Substantially improved productivity was achieved, compared to previous work (5-HMFA yield 64.2%, based on *D*-fructose) [1]. A sustainable strategy for synthesis of 5-HMFA from biobased *D*-fructose was developed in a one-pot manner, which avoided the separation of intermediates and diminished equipment input.

## 3. Materials and Methods

### 3.1. Enzymes, Chemical and Materials

Betaine (B), glycerol (Gly), *D*-fructose, tetraethyl orthosilicate, 5-hydroxymethylfurfural (5-HMF), 5-(hydroxymethyl)furfurylamine (5-HMFA), *D*-alanine (*D*-Ala) and other reagents were obtained from Changzhou Runyou Chemicals Co. (Changzhou, China).

### 3.2. Synthesis of DES B:Gly and Solid Acid SG(SiO_2_)

DES B:Gly was synthesized by heating. Hydrogen-bond-donor glycerol (Gly) and hydrogen-bond-acceptor betaine (B) were well blended in a designed molar ratio (1:2, mol:mol) in an oil bath. After being incubated at 353.15 K by stirring (300 rpm) for 3 h, the formed homogeneous DES solution (betaine:glycerol, B:Gly) was collected. Solid acid catalyst SG(SiO_2_) was prepared as reported previously [20].

### 3.3. Procedure for Conversion of D-Fructose

*D*-Fructose (0.2–2.0 M) and 50 mL B:Gly–water solvent (B:Gly, 0–25 wt%) were blended into a stainless-steel autoclave (100-mL) with SG(SiO_2_) (0–5 wt%) catalysts under agitation (500 rpm) at 140–160 °C after 0.25–3.0 h. Prior to HPLC analysis, the obtained 5-HMF samples were diluted and filtered by means of a 0.22 μm syringe filter.

### 3.4. Recombinant E. coli AT and Its Biotransamination Conditions

Recombinant *E. coli* AT was employed to aminate 5-HMF into 5-HMFA. Cell culture and harvest were carried out as previously reported [23].

To test amine donor on the the influence of 5-HMF bio-transamination, various molar ratio of *D*-Ala-to-5-HMF (2:1–20:1) were loaded in 50 mL reaction medium (100 mM K_2_HPO_4_–KH_2_PO_4_ buffer, pH 7.5) containing B:Gly (20 wt%), 5-HMF (200 mM), and AT cells (0.050 g/mL, wet weight) at 35 °C for 24 h. In order to test the substrate 5-HMF tolerance of AT cells in B:Gly–water medium, 50–500 mM commercial 5-HMF was separately blended with AT cells (0.050 g/mL, wet weight) and *D*-Ala (*D*-Ala-to-5-HMF molar ratio 15:1) in 50 mL reaction medium (100 mM K_2_HPO_4_–KH_2_PO_4_ buffer, pH 7.5) containing B:Gly (20 wt%) at 35 °C for 24 h.

### 3.5. A Hybrid Conversion of D-Fructose from 5-HMFA

Amounts of 200 mM *D*-fructose and 50 mL B:Gly–water solvent (B:Gly, 20 wt%) were blended into a stainless-steel autoclave (100 mL) with SG(SiO_2_) (3 wt%) catalysts by stirring (500 rmp) at 150 °C for 1.5 h. The resulting 5-HMF liquid was adjusted to pH 7.5 and, then, *E. coli* AT (0.050 g/mL, wet weight) and *D*-Ala (*D*-Ala-to-5-HMF molar ratio 15:1) were added to the bio-transamination system to generate 5-HMFA at 35 °C for 24 h.

### 3.6. Analytical Methods

Prior to HPLC analysis, the sample solution was passed through a 0.22-μm milli-pore filter. 5-HMF, 5-HMFA, and BHMF were analyzed on SHIMADZU LC-2030C HPLC system equipped with a Discovery^®^ C18 (4.6 mm × 250 mm, 5 μm) column (Appendix A). Mixtures containing 20 v% methanol and 80 v% H_2_O containing 0.1% trifluoroacetic acid were utilized as the mobile phase. Column temperature was kept at 35 °C, and the flow rate of the mobile phase was maintained at 0.8 mL/min. 5-HMFA and BHMF were monitored at 210 nm. 5-HMF was monitored at 254 nm.

The 5-HMF yield, 5-HMFA yield, BHMF yield and 5-HMFA selectivity were calculated as follows:(1)$$\mathrm{Yield}\;\mathrm{of}\;5-\mathrm{HMFA}=\frac{5-\mathrm{HMFA}\;\mathrm{produced}\;(\mathrm{mM})}{\mathrm{Initial}\;5-\mathrm{HMF}\;(\mathrm{mM})}\times 100\%$$
(2)$$\mathrm{Yield}\;\mathrm{of}\;5-\mathrm{HMF}=\frac{5-\mathrm{HMF}\;\mathrm{produced}\;(\mathrm{mM})}{\mathrm{Initial}\;D-\mathrm{fructose}\;(\mathrm{mM})}\times 100\%$$
(3)$$\mathrm{Yield}\;\mathrm{of}\;\mathrm{BHMF}=\frac{\mathrm{BHMF}\;\mathrm{produced}\;(\mathrm{mM})}{\mathrm{Initial}\;5-\mathrm{HMF}\;(\mathrm{mM})}\times 100\%$$
(4)$$\mathrm{Selectivity}\;\mathrm{of}\;5-\mathrm{HMFA}=\frac{5-\mathrm{HMFA}\;\mathrm{produced}\;(\mathrm{mM})}{\left(5-\mathrm{HMFA}+\mathrm{BHMF})\;\mathrm{produced}\;(\mathrm{mM}\right)}\times 100\%$$

## 4. Conclusions

As an important 5-HMF derivative, 5-HMFA is widely used in the production of diuretics, antihypertensive drugs, preservatives and curing agents. In this work, SG(SiO_2_), as a chemo-catalyst, was employed to catalyze *D*-fructose into 5-HMF at 150 °C within 90 min in a B:Gly–water system, achieving the maximum 5-HMF yield (91.7%, based on *D*-fructose). *E. coli* AT catalyzed 5-HMF into 5-HMFA at 35 °C, reaching a remarkable 5-HMFA yield of 94.1% (based on 5-HMF). This was the first example of the efficiently chemoenzymatic synthesis of 5-HMFA from biobased *D*-fructose by a hybrid catalysis with SG(SiO_2_) chemo-catalyst and *E. coli* AT cell bio-catalyst in B:Gly–water, indicating that this strategy has great potential for the manufacturing of 5-HMFA from *D*-fructose under green mild reaction conditions.

## Figures and Tables

**Figure 1 molecules-27-05748-f001:**
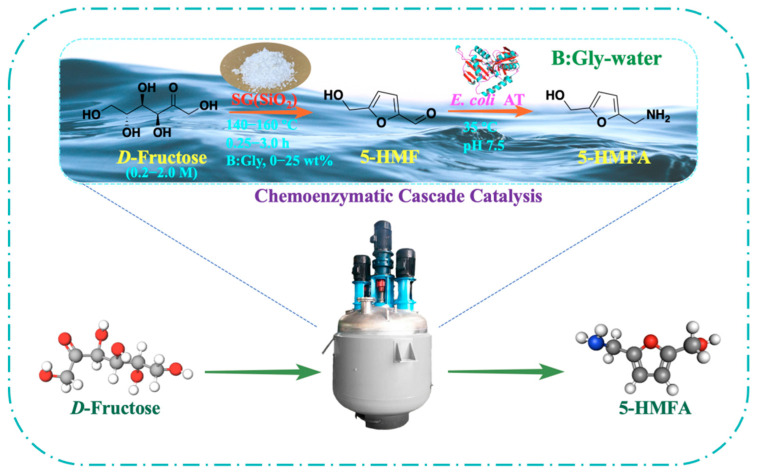
A hybrid synthetic route for the chemoenzymatic sustainable synthesis of 5-HMFA from *D*-fructose in betaine-based DESs–water medium.

**Figure 2 molecules-27-05748-f002:**
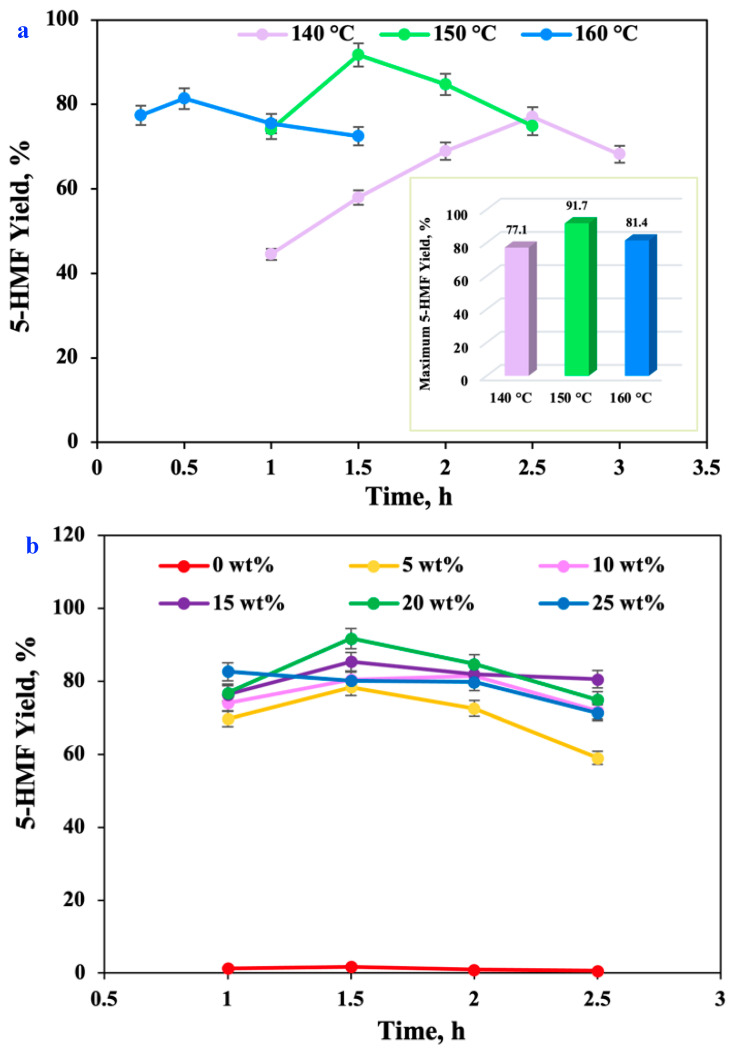
Effect of solvent B:Gly composition (0–25 wt%) on the 5-HMF formation [SG(SiO_2_) 3 wt%, 150 °C, 1–2.5 h] (**a**); Effect of reaction duration (0.25–3 h) and reaction temperature (140–160 °C) on the 5-HMF formation [B:Gly 20 wt%, SG(SiO_2_) 3 wt%] (**b**).

**Figure 3 molecules-27-05748-f003:**
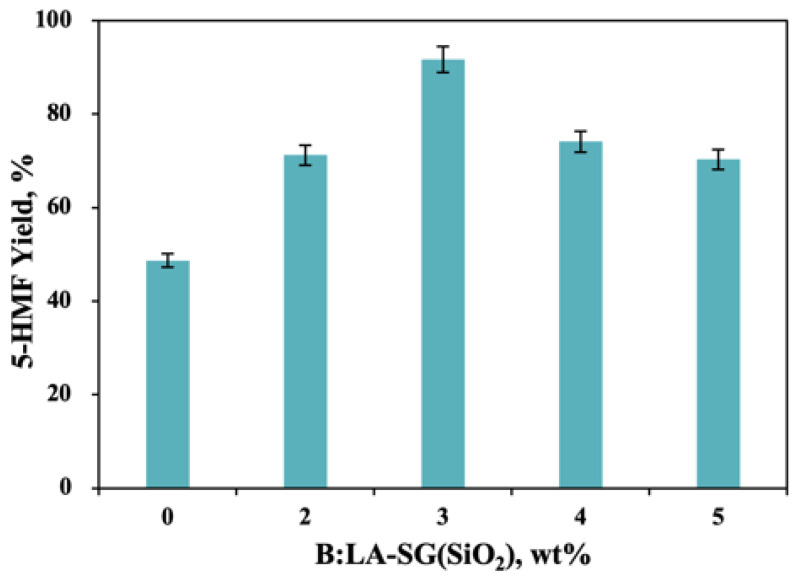
Effect of solid acid dose (SG(SiO_2_) 0–5 wt%) on the 5-HMF formation [B:Gly 20 wt%, 150 °C, 1.5 h].

**Figure 4 molecules-27-05748-f004:**
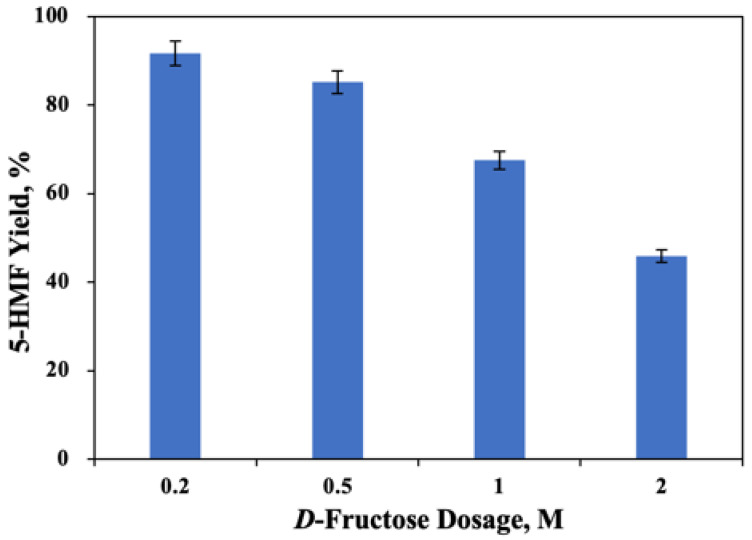
Effect of initial *D*-fructose dose (0.2–2 M) on the 5-HMF formation [B:Gly 20 wt%, SG(SiO_2_) 3 wt%, 150 °C, 1.5 h].

**Figure 5 molecules-27-05748-f005:**
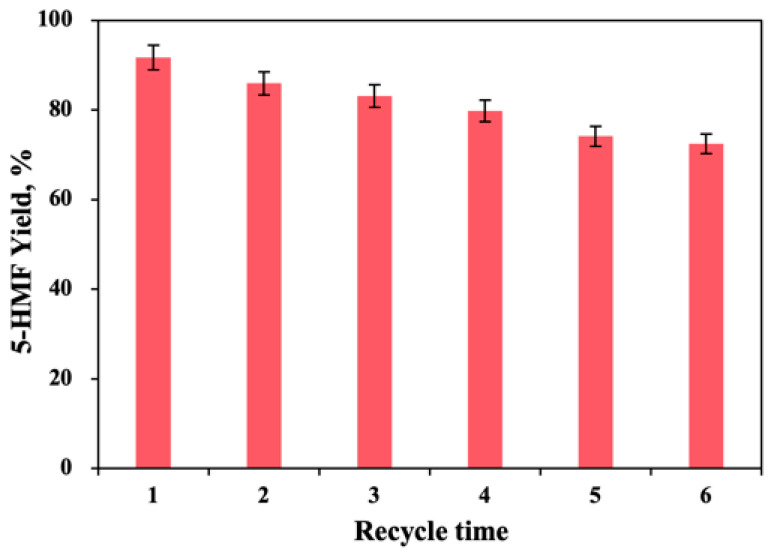
The reuse of the catalytic system on the 5-HMF generation.

**Figure 6 molecules-27-05748-f006:**
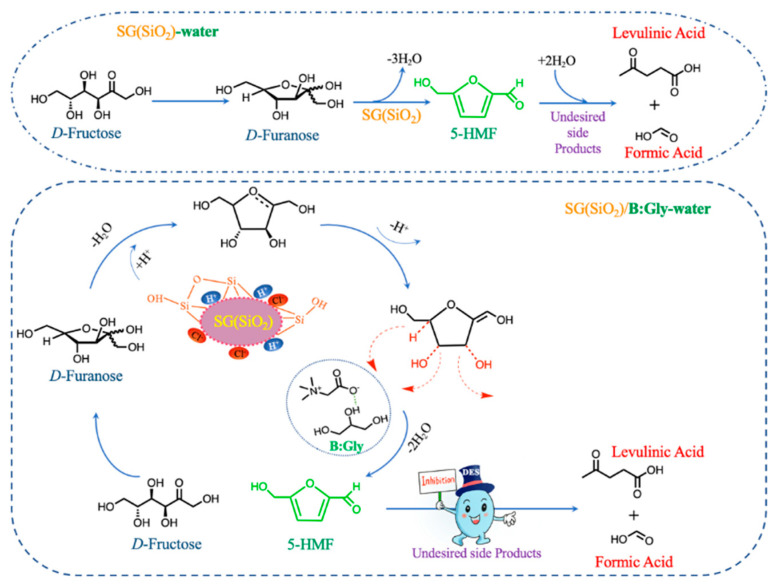
Possible mechanism for the dehydration of *D*-fructose.

**Figure 7 molecules-27-05748-f007:**
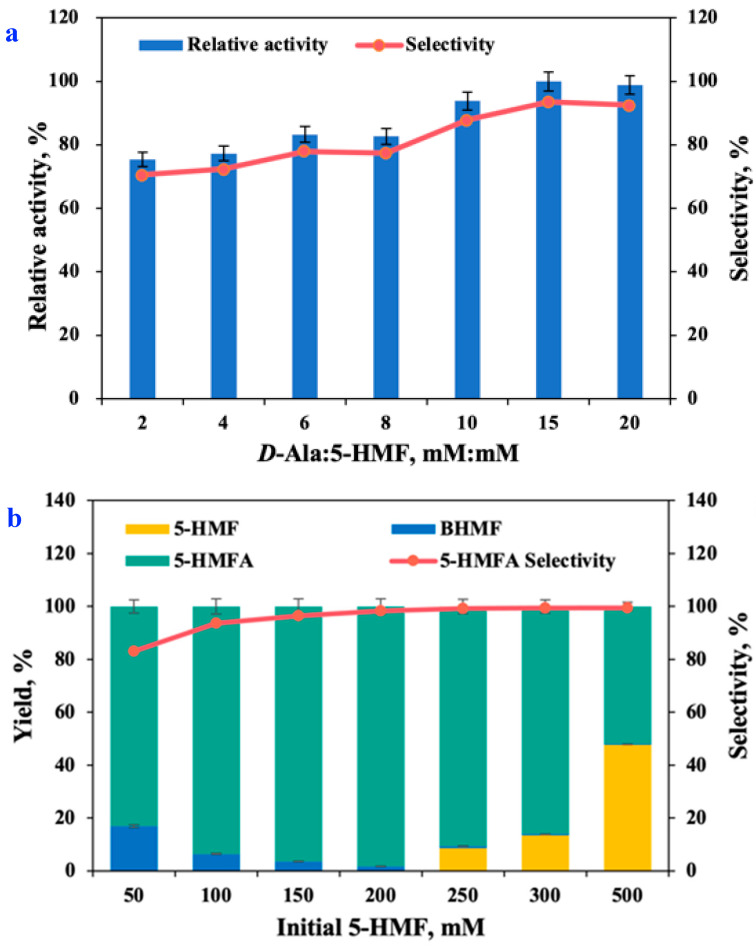
Effects of amine donor *D*-Ala dose (2:1–20:1 of *D*-Ala-to-5-HMF molar ratio) on the yield of 5-HMFA and selectivity [*E. coli* AT 0.050 g/mL, 35 °C, pH 7.5] (**a**); Effects of initial 5-HMF dose (50–500 mM) on the 5-HMFA yield and selectivity [*E. coli* AT 0.050 g/mL, 35 °C, pH 7.5, 15:1 of *D*-Ala-to-5-HMF molar ratio] (**b**).

**Figure 8 molecules-27-05748-f008:**
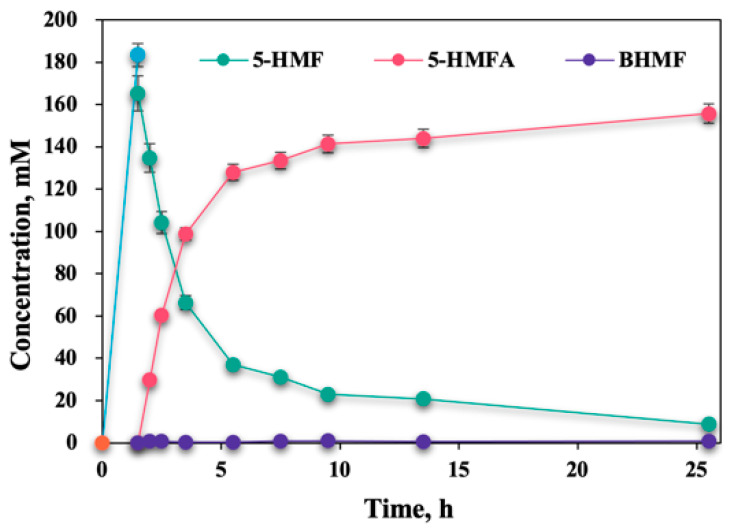
Time courses for chemoenzymatic transformation of *D*-fructose into 5-HMFA.

**Figure 9 molecules-27-05748-f009:**
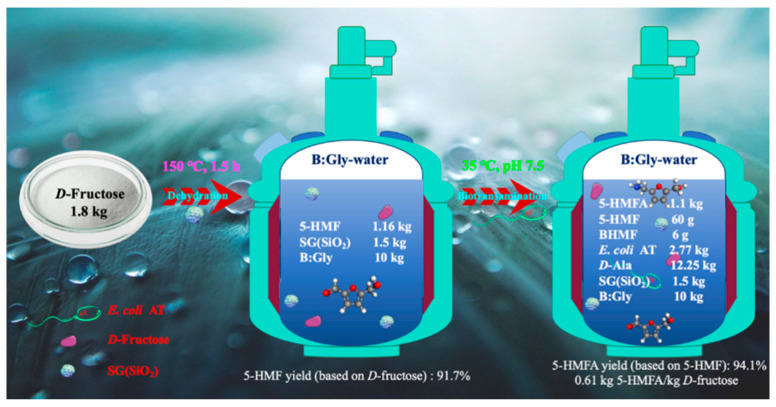
Mass balance from biobased *D*-fructose to 5-HMFA in B:Gly-water.

**Figure 10 molecules-27-05748-f010:**
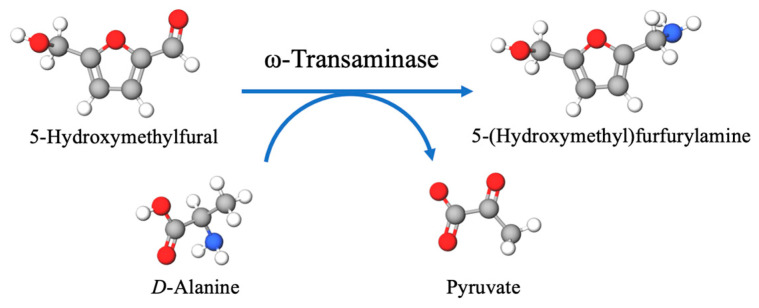
Bioamination of 5-HMF to 5-HMFA.

## Data Availability

Not applicable.

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
