# Peer review of "Significant Enhancement of 5-Hydroxymethylfural Productivity from D-Fructose with SG(SiO2) in Betaine:Glycerol–Water for Efficient Synthesis of Biobased 5-(Hydroxymethyl)furfurylamine"

_molecules, 2022, doi:10.3390/molecules27185748_

Round 1

Reviewer 1 Report

The authors described two interesting procedures for the synthesis of 5-HMF from D-fructose catalyzed by SG(SiO2) under eco-friendly conditions, alongside the chemoenzymatic catalysis of D-fructose to 5-HMFA in B:Gly−water using E. coli AT. I consider that the manuscript meets all requirements to be published in “Molecules” after minor revision. I did not find the supplementary material, including chromatograms that confirm the results in terms of yield, selectivity, and conversion. From an organic point of view, a plausible mechanism should be included to understand how is formed 5-MHFA. It would be very nice to confirm the formation of 5-HMF and 5-HMFA by NMR and HRMS techniques; however, HPLC chromatograms can confirm the formation of both products. Additional suggestions and comments are included:   (1) See Figure 1. The resolution should be improved.   (2) See 2.1. Investigation of B:Gly−water composition on the 5-HMF formation. (a) It is very important to explain how you determine the yield?, (b) authors do not mention results in terms of selectivity and conversion, and (c) all chromatograms to support the data in Figure 2 should be included in the Supplementary Material. (3) See Figure 2. It looks very crowded. Thus, the resolution should be improved. The size should be increased to see with better clarity. I consider that bar charts can be separated from Figure 2. (4) See 2.2. Effects of dehydration temperature and duration on the g 5-HMF. (a) Authors do not mention results in terms of selectivity and conversion ¿did you observe a by-product?, and (b) all chromatograms to support the data in Figure 2 should be included in the Supplementary Material. (5) See 2.3. Investigation of SG(SiO2) dose on the 5-HMF formation. (a) Authors do not mention results in terms of selectivity and conversion ¿did you observe a by-product?, and (b) all chromatograms to support the data in Figure 3 should be included in the Supplementary Material. (6) See 2.4. Investigation of D-fructose dosage on the 5-HMF formation. (a) Authors do not mention results in terms of selectivity and conversion ¿do you observe by-products like formic acid and levulinic acid?, and (b) all chromatograms to support the data in Figure 4 should be included in the Supplementary Material. (7) See 2.5. The reuse of the catalytic system. (a) Authors do not mention results in terms of selectivity and conversion ¿do you observe a by-product?, and (b) all chromatograms to support the data in Figure 5 should be included in the Supplementary Material. (8) See lines 177-189. It is very important to explain ¿how you experimentally recuperate the catalytic system?. (9) See lines 177-189. ¿Do you study the stability of the catalyst SG(SiO2)? It is very important to take an infrared spectrum from the 0, 3rd, and 6th batch. It should be included and explained in the manuscript.    (10) See 2.7. Biotransformation of 5-HMF into 5-HMFA. All chromatograms to support the data in Figure 7 should be included in the Supplementary Material. (11) See 2.8. Chemoenzymatic catalysis of D-fructose to 5-HMFA in B:Gly−water. All chromatograms to support the data in Figure 8 should be included in the Supplementary Material. (12) See 2.7. Biotransformation of 5-HMF into 5-HMFA. A plausible mechanism should be included to understand how is incorporated the amino group. (13) See references. The doi of each article should be added.

Author Response

Response Letter

Reviewer 1 #:

The authors described two interesting procedures for the synthesis of 5-HMF from D-fructose catalyzed by SG(SiO2) under eco-friendly conditions, alongside the chemoenzymatic catalysis of D-fructose to 5-HMFA in B:Gly−water using E. coli AT. I consider that the manuscript meets all requirements to be published in “Molecules” after minor revision. I did not find the supplementary material, including chromatograms that confirm the results in terms of yield, selectivity, and conversion. From an organic point of view, a plausible mechanism should be included to understand how is formed 5-MHFA.

It would be very nice to confirm the formation of 5-HMF and 5-HMFA by NMR and HRMS techniques; however, HPLC chromatograms can confirm the formation of both products.

Response: Thanks for the good suggestion. The  1H NMR of 5-HMFA was given:  1H NMR [(CD3OD, 400 MHz): δ 1.89 (s, 2H, NH2), 3.94 (s, 2H, CH2NH2), 4.50 (s, 2H, CH2OH), 6.27–6.28 (d,J = 4.0 Hz, 1H, furan H), 6.32–6.33 (d,J 4.0 Hz, 1H, furan H)]. However, due to COVID-19 and summer holiday, other analysis cannot be carried out.

Additional suggestions and comments are included:

(1) See Figure 1. The resolution should be improved.

Response: Thanks for the good suggestion. Figure 1 was improved.

  • See 2.1. Investigation of B:Gly−water composition on the 5-HMF formation.

(a) It is very important to explain how you determine the yield?,

Response: Thanks for the good suggestion. Prior to HPLC analysis, the sample solution was passed through a 0.22-μm millipore filter. 5-HMF, 5-HMFA, and BHMF were analyzed on SHIMADZU LC-2030C HPLC system equipped with a Discovery® C18 (4.6 mm× 250 mm, 5 μm) column. The mixture containing 20 v% methanol and 80 v% H2O containing 0.1 wt% trifluoroacetic acid were utilized as the mobile phase. Column temperature was kept at 35 °C, and the flow rate of the mobile phase was maintained at 0.8 mL/min. 5-HMFA and BHMF were monitored at 210 nm. 5-HMF was monitored at 254 nm.

The 5-HMF yield, 5-HMFA yield, and BHMF yield were calculated as related equations:

(b) authors do not mention results in terms of selectivity and conversion,

Response: Thanks for the good suggestion. These results and discussions were given in the revised manuscript. They were provided as below:

Figure 7a showed the effect of D-Ala dosages on the biotransamination. By increasing the D-Ala to 5-HMF molar ratio from 2:1 to 15:1 at 35 °C and pH 7.5 in B:Gly−water (B:Gly, 20 wt%), the biotransamination activity and 5-HMFA selectivity were increased. D-Ala and 5-HMF molar ratio of 15:1 was found to be the optimal reaction conditions. Further increasing the dosages of D-Ala in the reaction led to a slight decrease in the biocatalytic activity and 5-HMFA selectivity, indicating that potential inhibition occurred when the molar ratio of D-Ala-to-5-HMF exceeded 15:1 in the biotransamination reaction.

Hence, the substrate tolerance of recombinant E. coli AT was investigated under seven different dose of 5-HMF in B:Gly−water (Figure 7b). Interestingly, the selectivity of the product 5-HMFA was low when the substrate 5-HMF dose was low. With the continuous increase of 5-HMF dose, the selectivity of 5-HMFA also increased. E. coli AT could still catalyze the generation of 5-HMFA with high efficiency as the 5-HMF dosage was further increased to 200 mM. By raising the concentration of 5-HMF from 50 mM to 200 mM, the yield of 5-HMFA increased gradually. When the concentration of 5-HMF increased from 200 mM to 500 mM, the yield of 5-HMFA dropped from 98.3% to 52.0%, while the selectivity of bioamination remained in high level.

and (c) all chromatograms to support the data in Figure 2 should be included in the Supplementary Material.

Response: Thanks for the good suggestion. Only three representative HPLC images about substrate and product were provided in Support Information (Figure S1).

(3) See Figure 2. It looks very crowded. Thus, the resolution should be improved. The size should be increased to see with better clarity. I consider that bar charts can be separated from Figure 2.

Response: Thanks for the good suggestion. Figure 2 was improved. The high resolution of Figure 2 was given as below:

Figure 2. Effect of solvent B:Gly composition ( 0−25 wt%) on the 5-HMF formation [SG(SiO2) 3 wt%, 150 oC, 1−2.5 h] (a); Effect of reaction duration (0.25−3 h) and reaction temperature (140−160 °C) on the 5-HMF formation [B:Gly 20 wt%, SG(SiO2) 3 wt%] (b).

  • See 2.2. Effects of dehydration temperature and duration on the g 5-HMF. (a) Authors do not mention results in terms of selectivity and conversion ¿did you observe a by-product?,

Response: Thanks for the good suggestion. After the dehydration for 1-2.5 h, the conversions were 100%. Potential byproducts were not determined. Thus, the selectivity was not provided in this work.

and (b) all chromatograms to support the data in Figure 2 should be included in the Supplementary Material.

Response: Thanks for the good suggestion. Only three representative HPLC images about substrate and product were provided in Support Information (Figure S1).

(5) See 2.3. Investigation of SG(SiO2) dose on the 5-HMF formation. (a) Authors do not mention results in terms of selectivity and conversion ¿did you observe a by-product?,

Response: Thanks for the good suggestion. Potential byproducts were not determined. Thus, the selectivity was not provided in this work. The selectivity and conversion were not mentioned in this manuscript.

and (b) all chromatograms to support the data in Figure 3 should be included in the Supplementary Material.

Response: Thanks for the good suggestion. Only three representative HPLC images about substrate and product were provided in Support Information (Figure S1).

(6) See 2.4. Investigation of D-fructose dosage on the 5-HMF formation. (a) Authors do not mention results in terms of selectivity and conversion ¿do you observe by-products like formic acid and levulinic acid?,

Response: Thanks for the good suggestion. Potential byproducts (e.g., formic acid and levulinic acid) were not determined. Thus, the selectivity was not provided in this work. The selectivity and conversion were not mentioned in this manuscript.

and (b) all chromatograms to support the data in Figure 4 should be included in the Supplementary Material.

Response: Thanks for the good suggestion. Only three representative HPLC images about substrate and product were provided in Support Information (Figure S1).

(7) See 2.5. The reuse of the catalytic system. (a) Authors do not mention results in terms of selectivity and conversion ¿do you observe a by-product?,

Response: Thanks for the good suggestion. Potential byproducts were not determined. Thus, the selectivity was not provided in this work. The selectivity and conversion were not mentioned in this manuscript.

and (b) all chromatograms to support the data in Figure 5 should be included in the Supplementary Material.

Response: Thanks for the good suggestion. Only three representative HPLC images about substrate and product were provided in Support Information (Figure S1).

(8) See lines 177-189. It is very important to explain ¿how you experimentally recuperate the catalytic system?

Response: Thanks for the good suggestion. This recovery and reuse performance was given as below:

The regeneration and reuse of the catalytic system can decrease the economic burden and reduce environmental pollution, which is the key to the industrial production of 5-HMF and has high application potential [30,31]. As presented in Figure 5, the catalytic system was recycled six times in order to test the recyclability of the catalytic system to catalyze the synthesis of 5-HMF from D-fructose. 5-HMF was extracted from 5-HMF liquor for four times using ethyl acetate by mixing in equal volume. The B:Gly−water medium containing solid acid SG(SiO2) was reused for further production of 5-HMF. After the first round, the 5-HMF yield in the catalytic system was 91.7% and while the three rounds of reaction were over, the 5-HMF yield in the catalytic system decreased slightly to 83.1%. When D-fructose was catalyzed from 3rd to 6th batch, 5-HMF yields decreased from 83.1% to 72.4%. This might be due to the inevitable accumulation of humic substances, the yield of 5-HMF began to decline. In the 6th batch, the catalytic system also efficiently catalyzed D-fructose into 5-HMF. Reuse of CrCl3-[N2222]Cl/EG catalytic system from 1st to 4th batch, 5-HMF yield dropped from 42% to 38% [32]. These results indicated that SG(SiO2) catalyst with B:Gly−water could be efficiently recycled for the D-fructose dehydration to produce 5-HMF.

(9) See lines 177-189. ¿Do you study the stability of the catalyst SG(SiO2)? It is very important to take an infrared spectrum from the 0, 3rd, and 6th batch. It should be included and explained in the manuscript.

Response: Thanks for the good suggestion. However, due to COVID-19 and summer holiday, experiments and tests cannot be carried out.

(10) See 2.7. Biotransformation of 5-HMF into 5-HMFA. All chromatograms to support the data in Figure 7 should be included in the Supplementary Material.

Response: Thanks for the good suggestion. Only three representative HPLC images about substrate and product were provided in Support Information.

(11) See 2.8. Chemoenzymatic catalysis of D-fructose to 5-HMFA in B:Gly−water. All chromatograms to support the data in Figure 8 should be included in the Supplementary Material.

Response: Thanks for the good suggestion. Only three representative HPLC images about substrate and product were provided in Support Information (Figure S1).

(12) See 2.7. Biotransformation of 5-HMF into 5-HMFA. A plausible mechanism should be included to understand how is incorporated the amino group.

Response: Thanks for the good suggestion. In the revised manuscript, A plausible mechanism was added to understand how is incorporated the amino group.

  1. coli AT cell harboring ω-transaminase was able to aminate D-fructose-derived 5-HMF into produce 5-HMFA by using D-Ala as amine donor under the ambient condition, accompanying with pyruvate (Figure 10)

Figure 10. Bioamination of 5-HMF to 5-HMFA.

  • See references. The doi of each article should be added.

Response: Thanks for the good suggestion. In the revised manuscript, the doi of each article was added.

Reviewer 2 Report

This manuscript deals with the production of 5-hydroxymethylfural and 5-(hydroxymethyl)furfurylamine from fructose through the combination of chemical and enzymatic catalysis. Satisfactory products yield could be achieved in the optimized reaction conditions. The reported chemoenzymatic cascade catalysis procedure for the production of value-added chemicals from fructose circumvents the separation and purification intermediates, which simplified the production process and reduced the production cost. Overall, the research topic is interesting and important. This manuscript is well organized and give us informative viewpoints in this field. However, the following issues should be addressed before publishing.

1.     It is interesting to note that the yield of HMF could be significantly increased from 1.7% to 69.7% after the incorporation of small amount of B:Gly DES (5 wt%). This reviewer actually is confused about this result. Firstly, the so-called DES (B:Gly)-water solvent system in this manuscript can not be regarded as DES and water independently after being diluted by 95 wt% water, in this light, the B:Gly-water solvent system is just a homogeneous aqueous solution containing 5 wt% betaine and glycerol. The DES is thorough destroyed after the dilution. Secondly, the role of DES for the promotion of HMF production from fructose is still unclear and more evidence should be supplemented to reveal the reason behind this.

2.     The authors claimed that the introduction of B:Gly affected the hydrogen-bond network of the reaction system and HMF was more stable in B:Gly−water system than in pure water. However, no experimental results support these conclusions.

3.     In entry 2, Table 1, HMF yield of 48.7% was also obtained without the SG(SiO2) catalyst, the question is which component in the reaction system act as the acid catalyst to catalyze the dehydration of fructose in this case. 

4.     In the recycling experiment, the authors should provide more details for the experiment procedure, such as how to separate HMF from the reaction system and recycle the reaction medium.  

Considering these facts and the requirement of molecules, this manuscript can be accepted in the journal after major revision. 

Author Response

Response Letter

Reviewer 2#

This manuscript deals with the production of 5-hydroxymethylfural and 5-(hydroxymethyl)furfurylamine from fructose through the combination of chemical and enzymatic catalysis. Satisfactory products yield could be achieved in the optimized reaction conditions. The reported chemoenzymatic cascade catalysis procedure for the production of value-added chemicals from fructose circumvents the separation and purification intermediates, which simplified the production process and reduced the production cost. Overall, the research topic is interesting and important. This manuscript is well organized and give us informative viewpoints in this field. However, the following issues should be addressed before publishing.

  1. It is interesting to note that the yield of HMF could be significantly increased from 1.7% to 69.7% after the incorporation of small amount of . This reviewer actually is confused about this result. Firstly, the so-called DES (B:Gly)-water solvent system in this manuscript can not be regarded as DES and water independently after being diluted by 95 wt% water, in this light, the B:Gly-water solvent system is just a homogeneous aqueous solution containing 5 wt% betaine and glycerol. The DES is thorough destroyed after the dilution. Secondly, the role of DES for the promotion of HMF production from fructose is still unclear and more evidence should be supplemented to reveal the reason behind this.

Response: Thanks for the good suggestion. In this manuscript, the appropriate dosage of DES B:Gly was 20 wt%, and the optimal loading of solid acid SG(SiO2) was 3 wt%.Hydrogen-bond-donor glycerol (Gly) and hydrogen-bond-acceptor betaine (B). Betaine is acidic in solution. Also, DES B:Gly is acidic in solution. Small dosage of B:Gly DES (5 wt%) or betaine can promote the 5-HMF formation.

  1. The authors claimed that the introduction of B:Gly affected the hydrogen-bond network of the reaction system and HMF was more stable in B:Gly−water system than in pure water. However, no experimental results support these conclusions.

Response: Thanks for the good suggestion. In this revised manuscript, “The main reason was that medium containing B:Gly affected the hydrogen-bond network, and cocatalysis with B:Gly and SG(SiO2) could efficiently catalyze the 5-HMF formation from D-fructose.” Was deleted.

  1. In entry 2, Table 1, HMF yield of 48.7% was also obtained without the SG(SiO2) catalyst, the question is which component in the reaction system act as the acid catalyst to catalyze the dehydration of fructose in this case.

Response: Thanks for the good suggestion. DES B:Gly was synthesized by heating. Hydrogen-bond-donor glycerol (Gly) and hydrogen-bond-acceptor betaine (B). Betaine is acidic in solution. Also, DES B:Gly is acidic in solution.

  1. In the recycling experiment, the authors should provide more details for the experiment procedure, such as how to separate HMF from the reaction system and recycle the reaction medium.

Response: Thanks for the good suggestion. In this revised manuscript, more details for the experiment procedure was added as below:.

The regeneration and reuse of the catalytic system can decrease the economic burden and reduce environmental pollution, which is the key to the industrial production of 5-HMF and has high application potential [30,31]. As presented in Figure 5, the catalytic system was recycled six times in order to test the recyclability of the catalytic system to catalyze the synthesis of 5-HMF from D-fructose. 5-HMF was extracted from 5-HMF liquor for four times using ethyl acetate by mixing in equal volume. The B:Gly−water medium containing solid acid SG(SiO2) was reused for further production of 5-HMF. After the first round, the 5-HMF yield in the catalytic system was 91.7% and while the three rounds of reaction were over, the 5-HMF yield in the catalytic system decreased slightly to 83.1%. When D-fructose was catalyzed from 3rd to 6th batch, 5-HMF yields decreased from 83.1% to 72.4%. This might be due to the inevitable accumulation of humic substances, the yield of 5-HMF began to decline. In the 6th batch, the catalytic system also efficiently catalyzed D-fructose into 5-HMF. Reuse of CrCl3-[N2222]Cl/EG catalytic system from 1st to 4th batch, 5-HMF yield dropped from 42% to 38% [32]. These results indicated that SG(SiO2) catalyst with B:Gly−water could be efficiently recycled for the D-fructose dehydration to produce 5-HMF.

Considering these facts and the requirement of molecules, this manuscript can be accepted in the journal after major revision.

Response: Thanks for the good suggestion. We have revised our manuscript based on the reviewers’ comments.

Reviewer 3 Report

This work provides an efficient chemoenzymatic strategy for converting HMF into 5-(Hydroxymethyl)furfurylamine. The topic is interesting. This manuscript is well organized. I think it can be accepted after minor revision.

Other comments:

1. In “Abstract”, the productivity of HMFA based on the D-fructose loading should be given.

2. “2.2. Effects of dehydration temperature and duration on the g 5-HMF” should be corrected to “2.2. Effects of dehydration temperature and duration on the 5-HMF formation”.

3. “2.5. The reuse of the catalytic system” should be corrected to “2.5. The recyclability of the catalytic system”

4. “Plausible reaction mechanism” need be corrected to “Proposed catalytic mechanism for dehydration of D-fructose”.

5. Table 1 could be deleted.

Author Response

Reviewer 3#

This work provides an efficient chemoenzymatic strategy for converting HMF into 5-(Hydroxymethyl)furfurylamine. The topic is interesting. This manuscript is well organized. I think it can be accepted after minor revision.

Other comments:

  1. In “Abstract”, the productivity of HMFA based on the D-fructose loading should be given.

Response: Thanks for the good suggestion. In the revised manuscript, the productivity of HMFA based on the D-fructose loading was given as below:

Abstract: 5-Hydroxymethyl-2-furfurylamine (5-HMFA) as an important 5-HMF derivative has been widely utilized in the manufacture of diuretics, antihypertensive drugs, preservatives and curing agents. In this work, an efficient chemoenzymatic route was constructed for producing 5-(hydroxymethyl)furfurylamine (5-HMFA) from biobased D-fructose in deep eutectic solvent Betaine:Glycerol−water. The introduction of Betaine:Glycerol could greatly promote the dehydration of D-fructose to 5-HMF and inhibit the secondary decomposition reactions of 5-HMF compared with single aqueous phase. D-Fructose (200 mM) could be catalyzed to 5-HMF (183.4 mM) at 91.7% yield by SG(SiO2) (3 wt%) after 90 min in Betaine:Glycerol (20 wt%) at 150 oC. E. coli AT exhibited excellent biotransamination activity to aminate 5-HMF into 5-HMFA at 35 °C and pH 7.5. After 24 h, D-fructose-derived 5-HMF (165.4 mM) was converted to 5-HMFA (155.7 mM) in 94.1% yield with D-Ala (D-Ala-to-5-HMF molar ratio 15:1) in Betaine:Glycerol (20 wt%) without removal of SG(SiO2), chieving the productivity of 0.61 g 5-HMFA/(g substrate D-fructose). Chemoenzymatic valorisation of D-fructose with SG(SiO2) and E. coli AT was established for sustainable production of 5-HMFA, which had potential application.

  1. “2.2. Effects of dehydration temperature and duration on the g 5-HMF” should be corrected to “2.2. Effects of dehydration temperature and duration on the 5-HMF formation”.

Response: Thanks for the good suggestion. “2.2. Effects of dehydration temperature and duration on the g 5-HMF” was corrected to “2.2. Effects of dehydration temperature and duration on the 5-HMF formation”.

  1. “2.5. The reuse of the catalytic system” should be corrected to “2.5. The recyclability of the catalytic system”

Response: Thanks for the good suggestion. In the revised manuscript, “2.5. The reuse of the catalytic system” was corrected to “2.5. The recyclability of the catalytic system

  1. “Plausible reaction mechanism” need be corrected to “Proposed catalytic mechanism for dehydration of D-fructose”.

Response: Thanks for the good suggestion. In the revised manuscript, “Plausible reaction mechanism” was corrected to “Proposed catalytic mechanism for dehydration of D-fructose”.

  1. Table 1 could be deleted.

Response: Thanks for the good suggestion. In this revised manuscript, Table 1 was deleted from the text body, and it given in the Support Information.

Round 2

Reviewer 2 Report

Authors have well addressed the reviewer's comments. I recommend this manuscript in current form can be accepted for publication.